# Rate Coefficients of the Reactions of Fluorine Atoms with H_2_S and SH over the Temperature Range 220–960 K

**DOI:** 10.3390/molecules27238365

**Published:** 2022-11-30

**Authors:** Yuri Bedjanian

**Affiliations:** Institut de Combustion, Aérothermique, Réactivité et Environnement (ICARE), CNRS, CEDEX 2, 45071 Orléans, France; yuri.bedjanian@cnrs-orleans.fr; Tel.: +33-238255474

**Keywords:** fluorine atom, hydrogen sulfide, H_2_S, SH, kinetics, rate coefficient

## Abstract

Reaction F + H_2_S→SH + HF (1) is an effective source of SH radicals and an important intermediate in atmospheric and combustion chemistry. We employed a discharge-flow, modulated molecular beam mass spectrometry technique to determine the rate coefficient of this reaction and that of the secondary one, F + SH→S + HF (2), at a total pressure of 2 Torr and in a wide temperature range 220–960 K. The rate coefficient of Reaction (1) was determined directly by monitoring consumption of F atoms under pseudo-first-order conditions in an excess of H_2_S. The rate coefficient of Reaction (2) was determined via monitoring the maximum concentration of the product of Reaction (1), SH radical, as a function of [H_2_S]. Both rate coefficients were found to be virtually independent of temperature in the entire temperature range of the study: *k*_1_ = (1.86 ± 0.28) × 10^−10^ and *k*_2_ = (2.0 ± 0.40) × 10^−10^ cm^3^ molecule^−1^ s^−1^. The kinetic data from the present study are compared with previous room temperature measurements.

## 1. Introduction

Reactions of atomic fluorine, as a rule, are extremely fast, with rate coefficients often approaching the bimolecular collision frequency [1]. The rapidity of the elementary reactions of the F atoms arouses, in addition to theoretical interest, a practical interest for these reactions, in particular for the generation of active species (atoms and radicals) in laboratory gas-phase kinetic studies. For example, reactions of atomic fluorine with H_2_O [2], H_2_O_2_ [3], CH_4_ [4], and HNO_3_ [5] are often used as a source of OH, HO_2_, CH_3_, and NO_3_ radicals, respectively. In the present work we report the results of an experimental study of the reactions of F atoms with hydrogen sulfide:F + H_2_S→SH + HF(1)

Reaction (1) is a convenient and effective source of SH radicals, an important intermediate in combustion [6] and atmospheric chemistry [7]. The experimental study of the fast elementary reactions of F atoms is challenging, in particular, due to the presence of a rapid secondary chemistry. The secondary reaction, occurring in the F + H_2_S chemical system,
F + SH→S + HF(2)
was also investigated as a part of this study.

The kinetic information available for Reactions (1) and (2) is extremely scarce. There are no data on the temperature dependence of the reaction rate coefficients. One theoretical study [8] and only three relative [9,10,11] and one absolute measurement [12] of the rate coefficient of Reaction (1) are available, all realized at room temperature. The reported room temperature values of the rate coefficient differ by a factor of nearly 1.5. For Reaction (2), there is only one previous measurement, also at room temperature [12].

The objective of the present work was to determine the rate coefficients of the title reactions over a wide temperature range, T = 220–960 K, in order to provide a new kinetic data set for use in laboratory studies and, potentially, in theoretical reaction rate calculations.

## 2. Experimental

Experiments have been carried out at total pressure of 2 Torr of Helium in a flow tube reactor, employed in the laminar flow regime (Reynolds number < 10), and combined with a modulated molecular beam quadrupole mass spectrometer for the detection of the gas phase species. The experimental setup has been used extensively in the past, in particular, to study the kinetics and products of the reaction involving atomic fluorine [4,5,13,14,15,16]. Briefly, it consists of the gas introduction vacuum lines, the flow tube reactor, and the differentially pumped stainless steel high-vacuum chamber which houses the quadrupole mass spectrometer (Balzers, QMG 420, Balzers Aktiengesellschaft, Liechtenstein). Gas phase molecules sampled from the flow reactor are modulated by a tuning-fork chopper (35 Hz), ionized through impact with high kinetic energy electrons (18–30 eV, in this study) emitted by the ion source of the mass spectrometer and detected using an electron multiplier. Subsequently, the mass spectrometric signals are filtered and amplified with a lock-in amplifier and recorded for further analysis.

Two different flow reactors were used depending on temperature of the kinetic measurements. A low-temperature flow reactor covered the temperature range 220–320 K and consisted of a Pyrex tube (45 cm in length, 2.4 cm i.d.) with an outer jacket through which a temperature-regulated fluid was circulated. To minimize wall-loss of active species (F atoms and SH radicals), the inner surface of the reactor as well as the mobile injector were coated with halocarbon wax. A high-temperature flow reactor (Figure 1), used over the temperature range 300–960 K, consisted of a quartz tube (45 cm in length, 2.5 cm i.d.), where the temperature was controlled with electrical heating elements [17]. The temperature in the reactor was measured with a K-type thermocouple positioned in the middle of the reactor in contact with its outer surface. A temperature gradient along the flow tube measured with a thermocouple inserted in the reactor through the movable injector was found to be less than 1%.

Fluorine atoms were produced by discharging trace amounts of F_2_ in He in a microwave cavity (microwave generator Microtron 2000, 75 W, 2450 MHz, Electro-Medical Supplies Ltd., Wantage Oxfordshire England). To reduce F atom reactions with a glass surface inside the microwave cavity, a ceramic (Al_2_O_3_) tube was inserted in this part of the injector. It was verified by mass spectrometry that more than 95% of F_2_ was dissociated in the microwave discharge. The fluorine atoms were detected either as FCl (FCl^+^, *m*/*z* = 54) or as FBr at *m*/*z* = 98/100 (FBr^+^), after being scavenged in rapid reactions with excess Cl_2_ or Br_2_, respectively, added at the end of the reactor 5 cm upstream of the sampling cone:F + Cl_2_→Cl + FCl(3)
*k*_3_ = 6.0 × 10^−11^ cm^3^ molecule^−1^ s^−1^ (T = 180–360 K) [18].
F + Br_2_→Br + FBr(4)
*k*_4_ = 1.28 × 10^−10^ cm^3^ molecule^−1^ s^−1^ (T = 299–940 K) [14].

Absolute concentrations of F atoms were determined through their titration in Reactions (3) and (4) from the consumed fraction of Cl_2_ and Br_2_ ([F] = Δ[Cl_2_] = [FCl] and [F] = Δ[Br_2_] = [FBr]), respectively.

A similar approach was used to monitor two other labile species, S atoms and SH radicals, detected as BrS (BrS^+^, *m*/*z* = 111/113) and BrSH at *m*/*z* = 112/114 (BrSH^+^), respectively:S + Br_2_→Br + BrS(5)
*k*_5_ = 9.5 × 10^−11^ cm^3^ molecule^−1^ s^−1^ (T = 298 K) [19].
SH + Br_2_→Br + BrSH(6)
*k*_6_ = 5.7 × 10^−11^ exp(160/T) cm^3^ molecule^−1^ s^−1^ (T = 273–373 K) [20].

Absolute calibration of the mass spectrometric signals of BrSH was carried out as follows. First, the F atoms were titrated with an excess of Br_2_ in the main reactor, which led to the formation of FBr ([FBr]_0_). Then, the same concentration of F atoms was titrated with a mixture of Br_2_ and H_2_S resulting in the formation of FBr ([FBr]) and SH in Reactions (4) and (1), respectively. In the presence of Br_2_, SH radicals are rapidly converted to BrSH ([BrSH]) according to Reaction (6). The absolute concentration of BrSH was determined as [BrSH] = [FBr]_0_ − [FBr]. This calibration procedure avoids possible complications due to the rapid self-reaction of SH radicals:SH + SH→S + H_2_S(7)
*k*_7_ = 1.2 × 10^−11^ cm^3^ molecule^−1^ s^−1^ (T = 298 K) [21,22].

In several experiments, SH was detected directly at its parent peak (SH^+^, *m*/*z* = 33). In this case, the SH signals should be corrected for the contribution of H_2_S at *m*/*z* = 33 due to the dissociative ionization of H_2_S in the ion source of the mass spectrometer. At an electron energy of 18 eV, the contribution of H_2_S at *m*/*z* = 33 was ≤4% of the intensity of its parent peak (H_2_S^+^, *m*/*z* = 34).

The purities of the gases used were as follows: Br_2_ > 99.99% (Aldrich); Cl_2_ (>99%; Ucar); F_2_, 5% in helium (Alphagaz); and H_2_S > 99.5% (Alphagaz). Absolute concentrations of all stable species (H_2_S, F_2_, Cl_2_, and Br_2_) were derived from the measured flow rates of their manometrically prepared gas mixtures. He (the carrier gas) was taken directly from a high-pressure tank and had stated purity better than 99.999% (Alphagaz).

## 3. Results and Discussion

### 3.1. Rate Coefficient of Reaction (1)

The rate coefficient of Reaction (1), *k*_1_, was measured in an absolute way under pseudo-first order conditions in an excess H_2_S over F atoms ([F]_0_ = (0.6–1.2) × 10^11^ molecule cm^−3^, [H_2_S]/[F]_0_ = 2–35). The consumption of the excess reactant, H_2_S, in Reaction (1) generally did not exceed a few percent, although reaching up to 20% in a few kinetic runs. In all cases, the average concentration of H_2_S along the reaction zone was used in the calculations. In this series of experiments, Cl_2_ ([Cl_2_] ≈ 4 × 10^13^ molecule cm^−3^) was added at the end of the reactor (Figure 1) and F atoms were detected as FCl. In fact, measurements of the very high rate coefficient of Reaction (1) required the use of low concentrations of F atoms and the sensitivity of the mass spectrometer towards FCl was much better than that of FBr. The concentration vs. time profiles of both F-atom and H_2_S were monitored by changing the position of the movable injector (Figure 1). The distance between the injector head and the Cl_2_ introduction point (5 cm upstream of the sampling cone) was converted into reaction time using the linear flow velocity (2190–2830 cm s^−1^) of the gas mixture in the reactor. Figure 2 shows typical data obtained in this manner. The linearity of the semilogarithmic plots in Figure 2 clearly illustrates that the F-atom decays are first order, [F] = [F]_0_ × exp(−*k*_1_′ × t), where *k*_1_′ = *k*_1_ × [H_2_S].

Examples of the second-order plots observed at different temperatures are shown in Figure 3.

A linear least-squares fit of these data at each temperature gave the bimolecular rate coefficient. A summary of the absolute measurements of *k*_1_ is given in Table 1. The combined uncertainty on *k*_1_ was estimated to be around 15% by adding in quadrature statistical error (≤2%) and those on the measurements of the absolute concentration of H_2_S (~10%), flows (3%), pressure (2%), and temperature (1%). It is important to note that the current measurements of *k*_1_ were not affected by the fast secondary Reaction (2), given that the values of *k*_1_ and *k*_2_ are close (see below) and that [SH] ≤ Δ[F] << [H_2_S] under experimental conditions of the measurements.

The present data for *k*_1_ are plotted as a function of temperature in Figure 4 together with previous room temperature measurements [9,10,11,12]. In the single direct measurement of *k*_1_, Schölne et al. [12] used an experimental setup similar to that of the present study and determined the rate coefficient of Reaction (1) from the kinetics of H_2_S consumption in an excess of F atoms, *k*_1_ (±1σ) = (1.28 ± 0.04) × 10^−10^ cm^3^ molecule^−1^ s^−1^. This value is lower by a factor of 1.45 than the current measurement. Schölne et al. [12] also performed several experiments with an excess of H_2_S relative to F atoms. The rate coefficient of Reaction (1), determined under these conditions from the kinetics of HF formation, was *k*_1_ (±1σ) = (1.7 ± 0.4) × 10^−10^ cm^3^ molecule^−1^ s^−1^, which agrees very well with the present measurements. Smith et al. [10], measuring the relative HF infrared emission intensities from the reactions of F atoms with different compounds, determined *k*_1_ relative to the rate coefficient of Reaction (8), *k*_1_/*k*_8_ = 2.0 ± 0.2:F + CH_4_→CH_3_ + HF(8)
*k*_8_ = 1.28 × 10^−10^ exp(−219/T) cm^3^ molecule^−1^ s^−1^ (T = 220–960 K) [4].

A much higher value for this ratio, *k*_1_/*k*_8_ = 3.2 ± 0.3, can be extracted from the work of Williams and Rowland [9], who studied a number of competitive reactions involving thermalized ^18^F atoms. Persky [11], monitoring the decays of H_2_S and CH_4_ in reactions with F atoms in a flow reactor, determined *k*_1_/*k*_8_ = 2.35 ± 0.05 (±2σ). The relative rate data placed on an absolute basis with recently updated *k*_8_ = 6.14 × 10^−11^ cm^3^ molecule^−1^ s^−1^ (T = 298 K) [4] provide the following values for *k*_1_: (1.23 ± 0.25) × 10^−10^ [10], (1.44 ± 0.30) × 10^−10^ [11], and (1.96 ± 0.40) × 10^−10^ cm^3^ molecule^−1^ s^−1^ [9] with 20% uncertainty including 15% uncertainty on the rate coefficient of the reference Reaction (8) [4]. The last two values of *k*_1_ agree with the present measurements within the experimental uncertainties.

The continuous line in Figure 4 represents an exponential fit to the present data: *k*_1_ = (1.83 ± 0.03) × 10^−10^ exp((7 ± 4)/T) cm^3^ molecule^−1^ s^−1^ with statistical 2σ uncertainties on the precision of the fit. The experimental data for *k*_1_ are also well described (dashed line in Figure 4) with a temperature-independent value of *k*_1_ = (1.86 ± 0.28) × 10^−10^ cm^3^ molecule^−1^ s^−1^. We estimate the rate coefficient to be accurate within 15% over the investigated temperature range 220–960 K.

### 3.2. Rate Coefficient of Reaction (2)

The rate coefficient of F-atom reaction with SH was determined relative to that of Reaction (1). Kinetics of H_2_S and SH were monitored in an excess of F atoms over H_2_S. An example of the observed kinetic runs is shown in Figure 5. The simulated concentration vs. time profiles shown in Figure 5 were obtained within a simple mechanism including Reactions (1), (2) and (7) and wall loss of the active species involved, F + H_2_S→SH + HF, *k*_1_ = 1.86 × 10^−10^ cm^3^ molecule^−1^ s^−1^ (this work); F + SH→S + HF, *k*_2_ = 1.90 × 10^−10^ cm^3^ molecule^−1^ s^−1^ (fit), SH + SH→S + H_2_S, *k*_7_ = 1.2 × 10^−11^ cm^3^ molecule^−1^ s^−1^ [21,22]:F + wall loss *k*_9_ = 20 s^−1^ (this work)(9)
SH + wall→loss *k*_10_ = 35 s^−1^ (this work)(10)
S + wall→loss *k*_11_ = 120 s^−1^ (fit)(11)

The wall losses of F atoms and SH radicals were measured directly, monitoring the loss of these species in the absence of other reactants in the reactor. When measuring *k*_10_, the concentration of SH radicals (formed in Reaction (1) in an excess of H_2_S) was relatively low (≈10^11^ molecule cm^−3^) in order to minimize the contribution of the SH + SH reaction. Values of *k*_10_ measured at T = 220–960 K were in the range (20–35) s^−1^ with no notable correlation with temperature.

The data in Figure 5 show that the reaction mechanism used for the simulation gives an adequate representation of the chemical processes occurring in the reactor. The absolute concentrations of S atoms (detected as SBr, see Section 2) were not measured in the study; therefore, the corresponding experimental points in Figure 5 were simply scaled to the simulated profile. In addition, a first-order S-atom loss rate of 120 s^−1^ was included in the mechanism to better match experimental data. In any case, the data on the S-atom are not involved in the determination of *k*_2_ and are shown only for the sake of completeness.

One can note that the kinetics of two reactants, F-atom and H_2_S, are well described with *k*_1_, determined above in an excess of H_2_S. The SH concentration, as expected, increases to a maximum and then drops due to the loss of SH in Reactions (2), (7) and (10). The dotted line in Figure 5 shows the SH profile calculated without taking into account the SH losses in its self-Reaction (7) and on the wall (10). The relative contribution of these reactions increases with the reaction time (as F-atom is consumed). At the maximum concentration of SH, the impact of these two reactions can be considered negligible (the error bar displayed at the top of the SH profile corresponds to 5%), even for the relatively high initial concentration of H_2_S used in this experiment. Hence, one can conclude that the maximum concentration of SH is determined with a sufficient degree of accuracy by only two processes, the formation of SH in Reaction (1) and the consumption in Reaction (2), with *k*_1_[H_2_S] = *k*_2_[SH]_max_ at the time when [SH] is maximum. We did not perform the above analysis at other temperatures. It was assumed that the aforementioned conclusion is valid over the entire temperature range of the study, since the wall loss of SH does not vary significantly with temperature and the temperature dependence of the SH + SH reaction (unknown) is expected to be weak.

Thus, *k*_2_ was determined relative to *k*_1_ from the ratio *k*_1_/*k*_2_ = [SH]_max_/[H_2_S]. Experiments consisted of monitoring the concentrations of SH and H_2_S when [SH] reached its maximum. Examples of the experimental data observed at three different temperatures with varied initial concentration of H_2_S are shown in Figure 6. The slopes of the straight lines in Figure 6 provide the *k*_1_/*k*_2_ ratios at respective temperatures. The experimental conditions and the results of these measurements are summarized in Table 2. The values of *k*_2_ presented in Table 2 were calculated using *k*_1_ = 1.86 × 10^−^^10^ cm^3^ molecule^−^^1^ s^−^^1^ (T = 220–960 K), determined in the present work.

Temperature dependence of *k*_2_ is shown in Figure 7. The only value of *k*_2_ available in the literature is that reported by Schölne et al. [12]. The authors determined *k*_2_ = (2.0 ± 0.4) × 10^−10^ cm^3^ molecule^−1^ s^−1^ at T = 298 K from modeling SH kinetics in F + H_2_S system under an excess of F atoms over H_2_S. Fitting the current experimental data with an exponential function (solid line in Figure 7) yields the following Arrhenius expression: *k*_2_ = (2.14 ± 0.09) × 10^−10^ exp(−(23 ± 14)/T) cm^3^ molecule^−1^ s^−1^ with 2σ uncertainties representing the precision of the fit. The measured rate coefficient is observed to be independent of temperature, with all measured *k*_2_-values well within their corresponding uncertainties, *k*_2_ = (2.0 ± 0.4) × 10^−10^ cm^3^ molecule^−1^ s^−1^, in the temperature range of the measurements, T = 220–960 K. This value of *k*_2_ is recommended from the present study with a conservative uncertainty of 20% (including the uncertainty on the rate coefficient of the reference reaction).

## 4. Conclusions

In this work, using a discharge-flow reactor combined with mass spectrometry, we have investigated the kinetics of the reaction of atomic fluorine with hydrogen sulfide in a wide temperature range, 220–960 K. The F + H_2_S reaction is an effective source of SH radical (important intermediate in atmospheric and combustion chemistry) in laboratory studies. The rate coefficient of this reaction measured for the first time as a function of temperature was found to be practically independent of temperature with the value of *k*_1_ = (1.86 ± 0.28) × 10^−10^ cm^3^ molecule^−1^ s^−1^ at T = 220–960 K. Similar results were observed for the fast secondary reaction F + SH occurring in the F + H_2_S chemical system: *k*_2_ = (2.0 ± 0.4) × 10^−10^ cm^3^ molecule^−1^ s^−1^ at the same temperature range. The rate coefficient data for F atom reactions with H_2_S and SH obtained for the first time in an extended temperature range provide an experimental dataset for use in laboratory studies and theoretical reaction rate calculations.

## Figures and Tables

**Figure 1 molecules-27-08365-f001:**
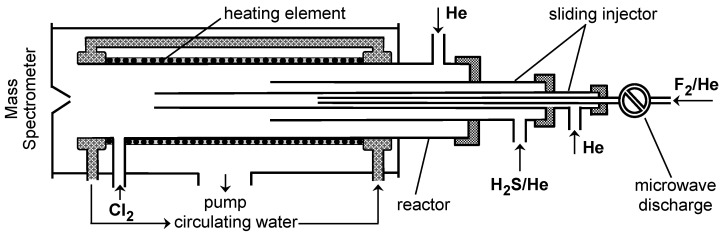
High-temperature flow reactor: configuration used in the measurements of the rate coefficient of reaction (1).

**Figure 2 molecules-27-08365-f002:**
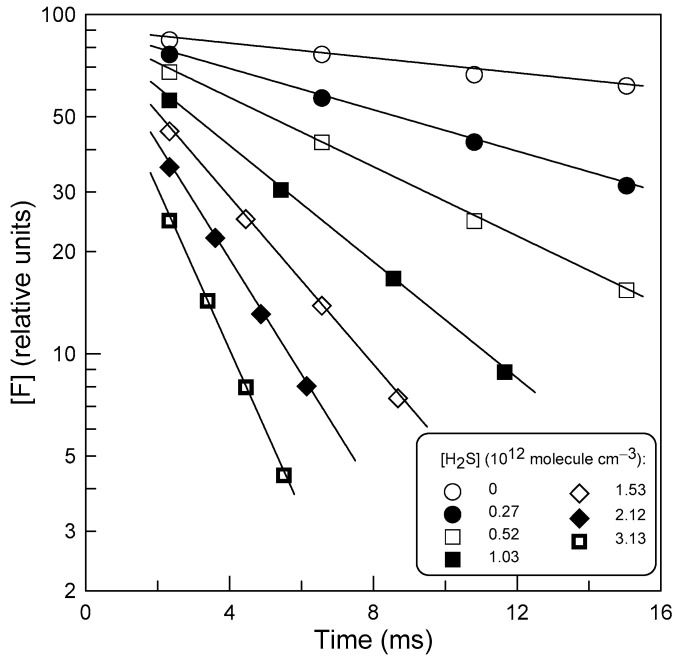
Typical F-atom decay profiles observed in the presence of different concentrations of H_2_S at *T* = 295 K.

**Figure 3 molecules-27-08365-f003:**
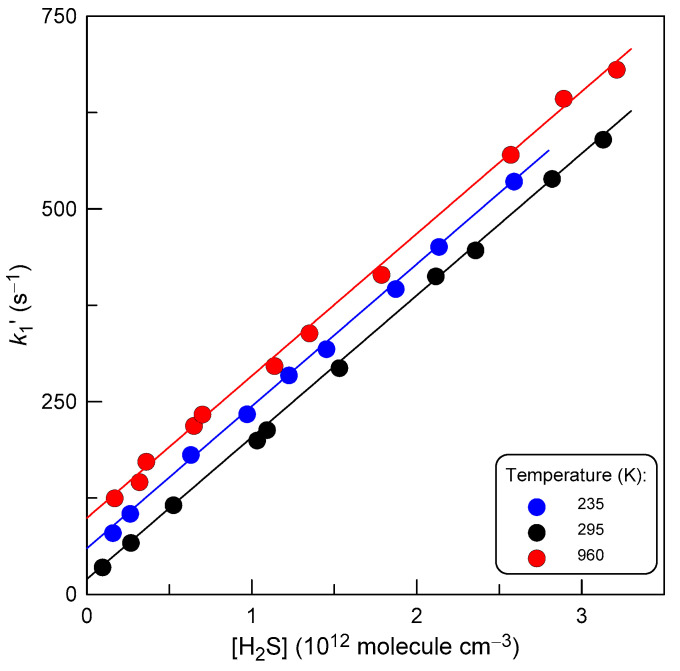
Dependence of the pseudo-first order rate constant, *k*_1_′ = *k*_1_[H_2_S], on the concentration of H_2_S at different temperatures. For clarity, *k*_1_′ data at T = 235 and 960 K are Y-shifted by 35 and 50 s^−1^, respectively.

**Figure 4 molecules-27-08365-f004:**
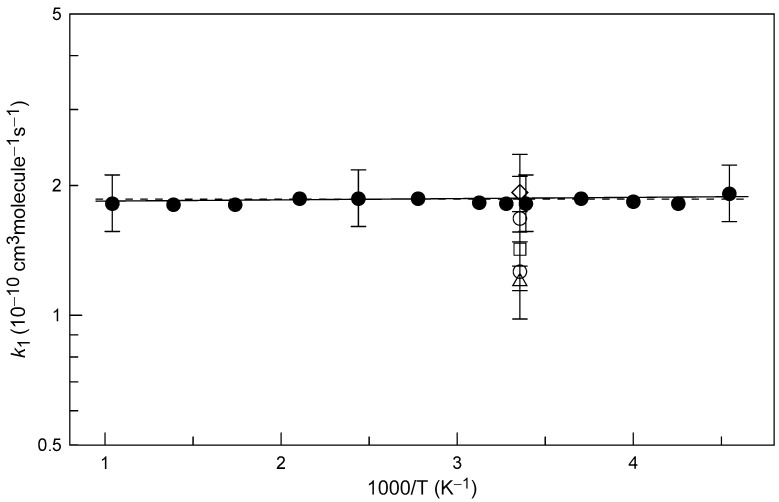
Summary of the measurements of *k*_1_: diamonds, Williams and Rowland [9]; triangles, Smith et al. [10]; open circles, Schölne et al. [12]; squares, Persky [11]; filled circles, this work. Uncertainties shown for selected present measurements of *k*_1_ correspond to 15%.

**Figure 5 molecules-27-08365-f005:**
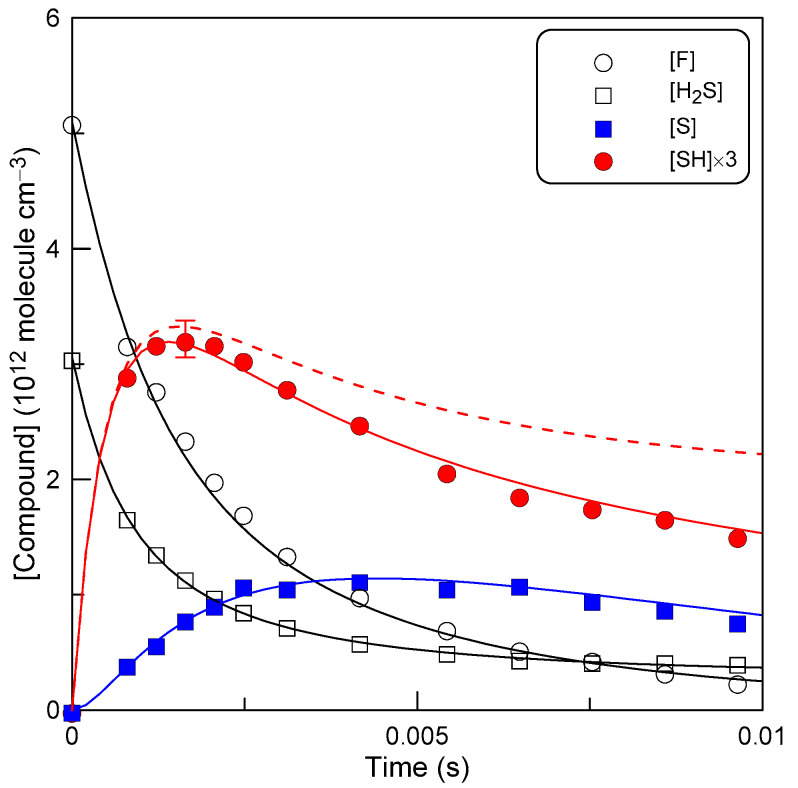
Reaction F + H_2_S: measured (points) and simulated (lines) kinetics of the reactants (F and H_2_S) and products (S and SH), T = 295 K.

**Figure 6 molecules-27-08365-f006:**
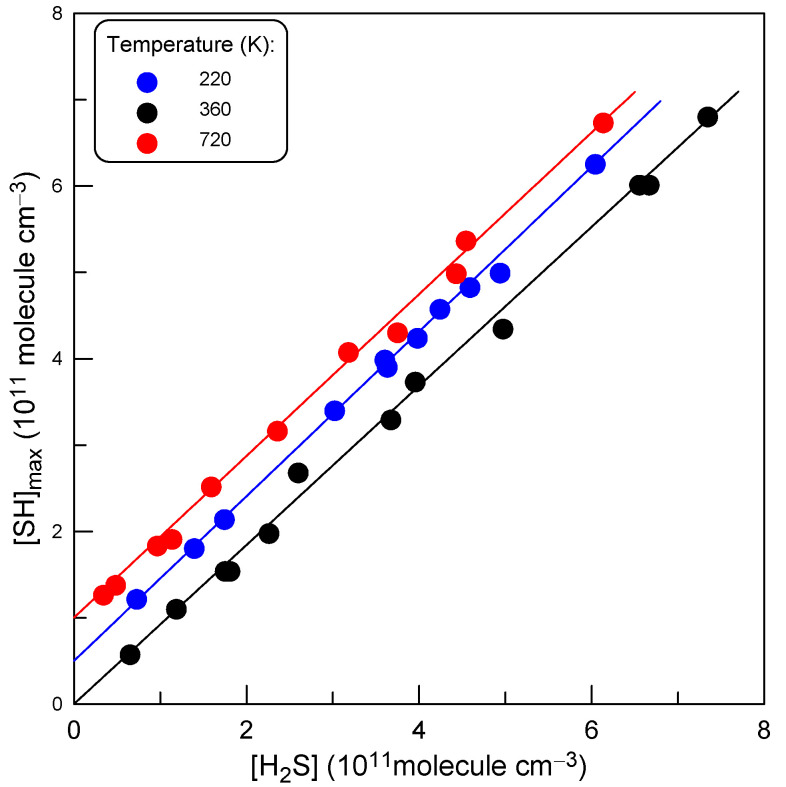
Typical dependence of [SH]_max_ on [H_2_S] (see text). For clarity, the experimental points at T = 220 and 720 K are Y-shifted by 0.5 and 1, respectively.

**Figure 7 molecules-27-08365-f007:**
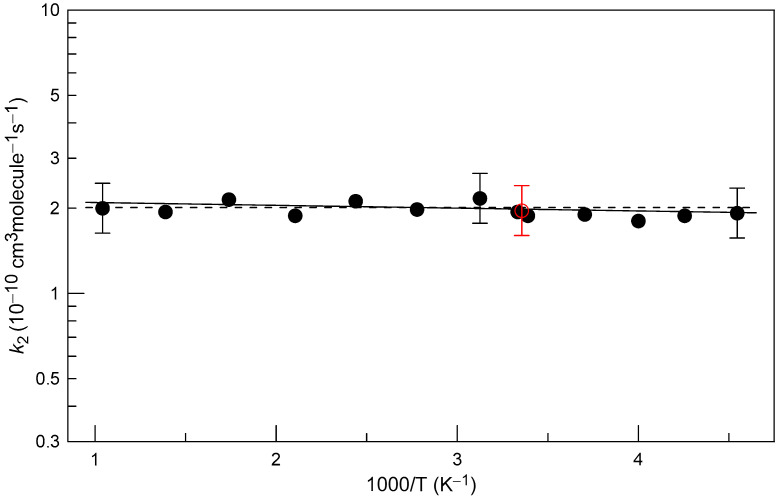
Temperature dependence of *k*_2_: open red circle, Schölne et al. [12]; filled circle, this work. Uncertainties shown for selected present measurements of *k*_2_ correspond to 20%.

**Table 1 molecules-27-08365-t001:** Experimental conditions and results of the measurements of the rate coefficient of reaction (1).

T (K) *^a^*	[H_2_S] *^b^*	*k*_1_ *^c^*	Reactor Surface *^d^*
220	0.11–2.87	1.94	HW
235	0.16–2.59	1.84	HW
250	0.19–2.26	1.86	HW
270	0.15–2.51	1.89	HW
295	0.27–3.13	1.84	HW
305	0.17–2.60	1.84	Q
320	0.13–2.95	1.85	HW
360	0.26–3.04	1.89	Q
410	0.16–2.03	1.89	Q
475	0.16–2.60	1.89	Q
575	0.14–4.44	1.83	Q
720	0.15–2.82	1.83	Q
960	0.17–3.21	1.84	Q

*^a^* 7–11 kinetic runs with different [H_2_S] at each temperature, [F]_0_ = (0.6–1.2) × 10^11^ cm^−3^. *^b^* Units of 10^12^ molecule cm^−3^; *^c^* units of 10^−10^ cm^3^ molecule^−1^ s^−1^; statistical 2σ uncertainty is ≤2%; total estimated uncertainty is 15%; *^d^* HW: halocarbon wax; Q: quartz.

**Table 2 molecules-27-08365-t002:** Experimental conditions and results of the measurements of the rate coefficient of Reaction (2).

T (K) *^a^*	[H_2_S]_0_ *^b^*	*k*_1_*/k*_2_ *^c^*	*k*_2_ *^d^*	Reactor Surface *^e^*
220	0.22–1.81	0.95	1.96	HW
235	0.12–2.23	0.97	1.92	HW
250	0.19–2.07	1.01	1.84	HW
270	0.13–2.80	0.96	1.94	HW
295	0.18–3.42	0.97	1.92	HW
300	0.22–2.73	0.94	1.98	Q
320	0.57–4.59	0.84	2.21	HW
360	0.19–2.21	0.92	2.02	Q
410	0.17–2.17	0.86	2.16	Q
475	0.15–2.05	0.97	1.92	Q
575	0.12–2.14	0.85	2.19	Q
720	0.10–1.84	0.94	1.98	Q
960	0.07–1.28	0.91	2.04	Q

*^a^* 8–12 measurements with different [H_2_S] at each temperature; *^b^* Units of 10^12^ molecule cm^−3^; *^c^* statistical 2σ uncertainty is ≤2.5%; *^d^* units of 10^−10^ cm^3^ molecule^−1^ s^−1^, total estimated uncertainty is 25%; *^e^* HW: halocarbon wax; Q: quartz.

## Data Availability

The data supporting reported results are available in this article.

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
