# Peer review of "Rate Coefficients of the Reactions of Fluorine Atoms with H2S and SH over the Temperature Range 220–960 K"

_molecules, 2022, doi:10.3390/molecules27238365_

Round 1

Reviewer 1 Report

The author presents his work on the measurement of the rate coefficients for (1) the F + H2S -> SH + HF and (2) F + SH -> S + HF reactions. The work is well presented and the scientific method is sound.

The following are my minor comments:

Abstract

Line 8. "... SH radical, ..." -> " ... SH radicals, ..."

Line 17. Please remove the word "few"

Experimental

General comments. How was the temperature measured and what was the temperature measurement accuracy; please list appropriate thermocouple references if appropriate. Same question regarding the pressure measurement within the reactor.

Throughout the text, I propose to use the word "coefficient" instead of "constant". Even though no temperature dependence has been observed in the rate coefficient measurements for the given reactions, temperature dependence was studied.

Line 47. Here, the author states that the experiments were carried out under "laminar flow regime". Please give the linear flow rate (cm/s).

Line 70. How were the F atoms generated? I assume that the F atoms were generated using a microwave cavity. Is so, please list experimental details such as F atom concentration within the cavity, cavity pressure, type of microwave cavity used, microwave generator reference ...

Results and Discussion

Line 107-108. The author states that the k1 value was measured under pseudo-first-order conditions. Please give [H2S] relative to [F]0 .... for example [H2S] approx 10[F]

Line 107. "absolute" -> "in situ" ?

Line 116. Given such a high linear flow rate (2190-2830 cm s-1) please state the Raynold's number to better support the experimental laminar flow conditions.

Table 1. (Page 5 of 10). Please list uncertainties is k1. Also, please add [F]0=(0.6-1.2)x1011 cm-3 in the legend.

Lines 170-173. The phrase is awkward. Please rewrite. I suggest to make two separate sentences.

Page 6 of 10. Equation (9). Two typos. Please correct.

Line 180. "The data in Figure 5 seems to show ..." -> "The date in Figure 5 shows ..."

Page 8 of 10. Table 2. Please list uncertainties in k2.

Line 222. The author states here "... under an excess of F atoms over H2S." Isn't it the other way around ? That is, H2S over F-atoms ?

Line 225. I would remove the word "practically" and add a statement that resembles something as "... the measured rate coefficient is observed to be independent of temperature, with all measured k2-values well within their corresponding uncertainties ... "

Conclusions

Line 236. Please remove the word "convenient" and correct the phrase correspondingly.

Line 240. "T = 220 -960 K" -> "T = 220 - 960 K" (add space after "-")

Author Response

Response to Reviewer 1 Comments

The author presents his work on the measurement of the rate coefficients for (1) the F + H2S -> SH + HF and (2) F + SH -> S + HF reactions. The work is well presented and the scientific method is sound.

The following are my minor comments:

Abstract

Line 8. "... SH radical, ..." -> " ... SH radicals, ..."

Corrected.

Line 17. Please remove the word "few"

Removed.

Experimental

General comments. How was the temperature measured and what was the temperature measurement accuracy; please list appropriate thermocouple references if appropriate. Same question regarding the pressure measurement within the reactor.

The following comment is added in the revised manuscript: “The temperature in the reactor was measured with a K-type thermocouple positioned in the middle of the reactor in contact with its outer surface. A temperature gradient along the flow tube measured with a thermocouple inserted in the reactor through the movable injector was found to be less than 1%.”

Throughout the text, I propose to use the word "coefficient" instead of "constant". Even though no temperature dependence has been observed in the rate coefficient measurements for the given reactions, temperature dependence was studied.

“constant” replaced with “coefficient” throughout the text.

Line 47. Here, the author states that the experiments were carried out under "laminar flow regime". Please give the linear flow rate (cm/s).

“Reynolds number < 10” is added in the text. Linear flow rates are presented in line 125.

Line 70. How were the F atoms generated? I assume that the F atoms were generated using a microwave cavity. Is so, please list experimental details such as F atom concentration within the cavity, cavity pressure, type of microwave cavity used, microwave generator reference ...

The sentence was modified as following: “Fluorine atoms were produced by discharging trace amounts of F2 in He in a microwave cavity (microwave generator Microtron 2000, 75 W, 2450 MHz).”

We are not able to measure the concentration of F atoms in the cavity, but it can be easily estimated from the concentration of F atoms in the reactor. In fact, from the point of view of kinetic measurements, it does not matter at all what happens in the microwave resonator. What is important is what happens in the reaction zone, that is, in the reactor.

Results and Discussion

Line 107-108. The author states that the k1 value was measured under pseudo-first-order conditions. Please give [H2S] relative to [F]0 .... for example [H2S] approx 10[F]0

The following text is added: “…[H2S]/[F]0 = 2 – 35). The consumption of the excess reactant, H2S, in reaction (1) generally did not exceed a few percent, although reaching up to 20 % in a few kinetic runs. In all cases, the average concentration of H2S along the reaction zone was used in the calcula-tions”

Line 107. "absolute" -> "in situ" ?

No, rather "directly". The term "absolute measurement" is used in kinetic studies to distinguish it from so-called relative rate method, where the rate coefficient is determined relative to that of a reference reaction.

Line 116. Given such a high linear flow rate (2190-2830 cm s-1) please state the Raynold's number to better support the experimental laminar flow conditions.

“Reynolds number < 10” is added in the text (line 48).

Table 1. (Page 5 of 10). Please list uncertainties is k1. Also, please add [F]0=(0.6-1.2)x1011 cm-3 in the legend.

Uncertainties are given in the legend: “statistical 2σ uncertainty is ≤ 2%; total estimated uncertainty is 15 %.” We do not consider it appropriate to detail in the table the statistical uncertainties of 1.5-2%. “[F]0=(0.6-1.2)x1011 cm-3” is added in the legend.

Lines 170-173. The phrase is awkward. Please rewrite. I suggest to make two separate sentences.

The phrase has been reworded into two separate sentences.

Page 6 of 10. Equation (9). Two typos. Please correct.

Corrected.

Line 180. "The data in Figure 5 seems to show ..." -> "The date in Figure 5 shows ..."

Corrected.

Page 8 of 10. Table 2. Please list uncertainties in k2.

Uncertainties are given in the legend: statistical 2σ uncertainty of ≤ 2.5% on k1/k2 and total estimated uncertainty of 25 % on k2. We do not consider it appropriate to list in the table the estimated total uncertainty of 25%, which is similar for all the temperatures.

Line 222. The author states here "... under an excess of F atoms over H2S." Isn't it the other way around ? That is, H2S over F-atoms ?

No, the statement is correct.

Line 225. I would remove the word "practically" and add a statement that resembles something as "... the measured rate coefficient is observed to be independent of temperature, with all measured k2-values well within their corresponding uncertainties ... "

The sentence is rephrased as suggested.

Conclusions

Line 236. Please remove the word "convenient" and correct the phrase correspondingly.

Corrected.

Line 240. "T = 220 -960 K" -> "T = 220 - 960 K" (add space after "-")

Done.

Reviewer 2 Report

The paper presents the first rate constant measurements for the reactions of H2S and SH with the fluorine atom over an extended temperature range. The experimental apparatus and methodology is adequately described in the paper, as well as the strategy to derive the rate constant of the secondary reaction. The data are of high interest for the community.

Nonetheless, some information is lacking in the paper. In the introduction, the theoretical study of Korweitz and Persky (Chem. Phys. Lett., 1999, Vol. 307, pp. 479-483) is worth mentioning. In the experimental section, the author should indicate the pressure at which the experimental measurements were carried out.

Regarding the secondary reaction SH + F, the author uses the data of Figure 5 and the proposed simple mechanism to justify that the SH-loss at the wall can be neglected in interpreting the speciation data and thus that the rate of the title reaction can be derived form the ratio of [H2S]/[SH] when [SH] reaches its maximum. In the proposed mechanism, the rate constant of loss at the wall of the different atoms and radical are said to be determined in the present work. However, I could not find any explanation on how these rate constants were determined. Do they depend on temperature, and do they impact the k2 values determined at higher temperatures?

Has the author collected data similar to the ones in Figure 5 for all the conditions investigated? If so, has he tried to determine the value of k2 by fitting the modeling prediction to the results and how this optimized value would differ from the one obtained from the ones presented in the paper?

Line 194 mentions a dashed line that cannot be seen in Figure 5. At line 204, the [H2S]max nomenclature is confusing at it is stands not for the maximum H2S concentration, but the H2S concentration at the time when [SH] is maximum.

Author Response

Response to Reviewer 2 Comments

The paper presents the first rate constant measurements for the reactions of H2S and SH with the fluorine atom over an extended temperature range. The experimental apparatus and methodology is adequately described in the paper, as well as the strategy to derive the rate constant of the secondary reaction. The data are of high interest for the community.

Nonetheless, some information is lacking in the paper. In the introduction, the theoretical study of Korweitz and Persky (Chem. Phys. Lett., 1999, Vol. 307, pp. 479-483) is worth mentioning.

The reference was added in the revised manuscript.

In the experimental section, the author should indicate the pressure at which the experimental measurements were carried out.

“at total pressure of 2 Torr of Helium” was added in the text (line 47).

Regarding the secondary reaction SH + F, the author uses the data of Figure 5 and the proposed simple mechanism to justify that the SH-loss at the wall can be neglected in interpreting the speciation data and thus that the rate of the title reaction can be derived form the ratio of [H2S]/[SH] when [SH] reaches its maximum. In the proposed mechanism, the rate constant of loss at the wall of the different atoms and radical are said to be determined in the present work. However, I could not find any explanation on how these rate constants were determined. Do they depend on temperature, and do they impact the k2 values determined at higher temperatures?

The following comment is added in the text (pages 188-193): “The wall losses of F atoms and SH radicals were measured directly, monitoring the loss of these species in the absence of other reactants in the reactor. When measuring k10, the concentration of SH radicals (formed in reaction (1) in an excess of H2S) was relatively low (» 1011 molecule cm-3) in order to minimize the contribution of the SH + SH reaction. Values of k10 measured at Т = 220 - 960 K were in the range (20-35) s-1 with no notable correlation with temperature.”

Has the author collected data similar to the ones in Figure 5 for all the conditions investigated? If so, has he tried to determine the value of k2 by fitting the modeling prediction to the results and how this optimized value would differ from the one obtained from the ones presented in the paper?

The following comment is added in the text (pages 216-220): “We did not perform the above analysis at other temperatures. It was assumed that the aforementioned conclusion is valid over the entire temperature range of the study, since the wall loss of SH does not vary significantly with temperature and the temperature dependence of the SH + SH reaction (unknown) is expected to be weak.

Line 194 mentions a dashed line that cannot be seen in Figure 5.

The line is well visible: “dashed” is replaced with “dotted” in the text.

At line 204, the [H2S]max nomenclature is confusing at it is stands not for the maximum H2S concentration, but the H2S concentration at the time when [SH] is maximum.

The text has been modified to avoid misunderstandings.